# Adaptive SVRG Methods under Error Bound Conditions with Unknown Growth Parameter

**Yi Xu[†], Qihang Lin[‡], Tianbao Yang[†]**
[†]Department of Computer Science, The University of Iowa, Iowa City, IA 52242, USA
[‡]Department of Management Sciences, The University of Iowa, Iowa City, IA 52242, USA
{yi-xu, qihang-lin, tianbao-yang}@uiowa.edu

## Abstract

Error bound, an inherent property of an optimization problem, has recently revived in the development of algorithms with improved global convergence without strong convexity. The most studied error bound is the quadratic error bound, which generalizes strong convexity and is satisfied by a large family of machine learning problems. Quadratic error bound have been leveraged to achieve linear convergence in many first-order methods including the stochastic variance reduced gradient (SVRG) method, which is one of the most important stochastic optimization methods in machine learning. However, the studies along this direction face the critical issue that the algorithms must depend on an unknown growth parameter (a generalization of strong convexity modulus) in the error bound. This parameter is difficult to estimate exactly and the algorithms choosing this parameter heuristically do not have theoretical convergence guarantee. To address this issue, we propose novel SVRG methods that automatically search for this unknown parameter on the fly of optimization while still obtain almost the same convergence rate as when this parameter is known. We also analyze the convergence property of SVRG methods under Hölderian error bound, which generalizes the quadratic error bound.

## 1 Introduction

Finite-sum optimization problems have broad applications in machine learning, including regression by minimizing the (regularized) empirical square losses and classification by minimizing the (regularized) empirical logistic losses. In this paper, we consider the following finite-sum problem:

$$\min_{x \in \Omega} F(x) \triangleq \frac{1}{n} \sum_{i=1}^{n} f_i(x) + \Psi(x), \tag{1}$$

where $f_i(x)$ is a continuously differential convex function whose gradient is Lipschitz continuous and $\Psi(x)$ is a proper, lower-semicontinuous convex function [24]. Traditional proximal gradient (PG) methods or accelerated proximal gradient (APG) methods for solving (1) become prohibited when the number of components $n$ is very large, which has spurred many studies on developing stochastic optimization algorithms with fast convergence [4, 8, 25, 1].

An important milestone among several others is the stochastic variance reduced gradient (SVRG) method [8] and its proximal variant [26]. Under the strong convexity of the objective function $F(x)$, linear convergence of SVRG and its proximal variant has been established. Many variations of SVRG have also been proposed [2, 1]. However, the key assumption of strong convexity limits the power of SVRG for many interesting problems in machine learning without strong convexity. For example, in regression with high-dimensional data one is usually interested in solving the least-squares regression with an $\ell_1$ norm regularization or constraint (known as the LASSO-type problem). A common practice for solving non-strongly convex finite-sum problems by SVRG is to add a small strongly convex regularizer (e.g., $\frac{\lambda}{2}\|x\|_2^2$) [26]. Recently, a variant of SVRG (named SVRG++ [2]) was

designed that can cope with non-strongly convex problems without adding the strongly convex term. However, these approaches only have sublinear convergence (e.g., requiring a $O(1/\epsilon)$ iteration complexity to achieve an $\epsilon$-optimal solution).

Promisingly, recent studies on optimization showed that leveraging the quadratic error bound (QEB) condition can open a new door to the linear convergence without strong convexity [9, 20, 6, 30, 5, 3]. The problem (1) obeys the QEB condition if the following holds:

$$\|x - x_*\|_2 \le c(F(x) - F(x_*))^{1/2}, \forall x \in \Omega, \tag{2}$$

where $x_*$ denotes the closest optimal solution to $x$ and $\Omega$ is usually a compact set. Indeed, the aforementioned LASSO-type problems satisfy the QEB condition. It is worth mentioning that the above condition (or similar conditions) has been explored extensively and has different names in the literature, e.g., the second-order growth condition, the weak strong convexity [20], essential strong convexity [13], restricted strong convexity [31], optimal strong convexity [13], semi-strong convexity [6]. Interestingly, [6, 9] have showed that SVRG can enjoy a linear convergence under the QEB condition. However, the issue is that SVRG requires to know the parameter $c$ (analogous to the strong convexity parameter) in the QEB for setting the number of iterations of inner loops, which is usually unknown and difficult to estimate. A naive trick for setting the number of iterations of inner loops to a certain multiplicative factor (e.g., 2) of the number of components $n$ is usually sub-optimal and worrisome because it may not be large enough for bad conditioned problems or it could be too large for good conditioned problems. In the former case, the algorithm may not converge as the theory indicates and in the latter case, too many iterations may be wasted for inner loops.

To address this issue, we develop a new variant of SVRG that embeds an efficient automatic search step for $c$ into the optimization. The challenge for developing such an adaptive variant of SVRG is that one needs to develop an appropriate machinery to check whether the current value of $c$ is large enough. One might be reminded of some restarting procedure for searching the unknown strong convexity parameter in APG methods [21, 11]. However, there are several differences that make the development of such a search scheme much more daunting for SVRG than for APG. The first difference is that, although SVRG has a lower per-iteration cost than APG, it also makes smaller progress towards the optimality after each iteration, which provides less information on the correctness of the current $c$. The second difference lies at that the SVRG is inherently stochastic, making the analysis for bounding the number of search steps much more difficult. To tackle this challenge, we propose to perform the proximal gradient updates occasionally at the reference points in SVRG where the full gradient is naturally computed. The normal of the proximal gradient provides a probabilistic "certificate" for checking whether the value of $c$ is large enough. We then provide a novel analysis to bound the expected number of search steps with a consideration that the probabilistic "certificate" might fail with some probability. The final result shows that the new variant of SVRG enjoys a linear convergence under the QEB condition with unknown $c$ and the corresponding complexity is only worse by a logarithmic factor than that in the setting where the parameter $c$ is assumed to be known.

Besides the QEB condition, we also consider more general error bound conditions (aka the Hölderian error bound (HEB) conditions [3]) whose definition is given below, and develop adaptive variants of SVRG under the HEB condition with $\theta \in (0, 1/2)$ to enjoy intermediate faster convergence rates than SVRG under only the smoothness assumption (e.g, SVRG++ [2]). It turns out that the adaptive variants of SVRG under HEB with $\theta < 1/2$ are simpler than that under the QEB.

**Definition 1** (Hölderian error bound (HEB)). *Problem (1) is said to satisfy a Hölderian error bound condition on a compact set $\Omega$ if there exist $\theta \in (0, 1/2]$ and $c > 0$ such that for any $x \in \Omega$*

$$\|x - x_*\|_2 \le c(F(x) - F_*)^{\theta}, \tag{3}$$

*where $x_*$ denotes the closest optimal solution to $x$.*

It is notable that the above inequality can always hold for $\theta = 0$ on a compact set $\Omega$. Therefore the discussion in the paper regarding the HEB condition also applies to the case $\theta = 0$. In addition, if a HEB condition with $\theta \in (1/2, 1]$ holds, we can always reduce it to the QEB condition provided that $F(x) - F_*$ is bounded over $\Omega$. However, we are not aware of any interesting examples of (1) for such cases. We defer several examples satisfying the HEB conditions with explicit $\theta \in (0, 1/2]$ in machine learning to Section 5. We refer the reader to [29, 28, 27, 14] for more examples.

## 2 Related work

The use of error bound conditions in optimization for deriving fast convergence dates back to [15, 16, 17], where the (local) error bound condition bounds the distance of a point in the local neighborhood of the optimal solution to the optimal set by a multiple of the norm of the proximal gradient at the point. Based on their local error bound condition, they have derived local linear convergence for descent methods (e.g., proximal gradient methods). Several recent works have established the same local error bound conditions for several interesting problems in machine learning [7, 32, 33].

Hölderian error bound (HEB) conditions have been studied extensively in variational analysis [10] and recently revived in optimization for developing fast convergence of optimization algorithms. Many studies have leveraged the QEB condition in place of strong convexity assumption to develop fast convergence (e.g., linear convergence) of many optimization algorithms (e.g., the gradient method [3], the proximal gradient method [5], the accelerated gradient method [20], coordinate descent methods [30], randomized coordinate descent methods [9, 18], subgradient methods [29, 27], primal-dual style of methods [28], and etc.). This work is closely related to several recent studies that have shown that SVRG methods can also enjoy linear convergence for finite-sum (composite) smooth optimization problems under the QEB condition [6, 9, 12]. However, these approach all require knowing the growth parameter in the QEB condition, which is unknown in many practical problems. It is worth mentioning that several recent studies have also noticed the similar issue in SVRG-type of methods that the strong convexity constant is unknown and suggested some practical heuristics for either stopping the inner iterations early or restarting the algorithm [2, 22, 19]. Nonetheless, no theoretical convergence guarantee is provided for the suggested heuristics.

Our work is also related to studies that focus on searching for unknown strong convexity parameter in accelerated proximal gradient (APG) methods [21, 11] but with striking differences as mentioned before. Recently, Liu & Yang [14] considered the HEB for composite smooth optimization problems and developed an adaptive restarting accelerated gradient method without knowing the $c$ constant in the HEB. As we argued before, their analysis can not be trivially extended to SVRG.

## 3 SVRG under the HEB condition in the oracle setting

In this section, we will present SVRG methods under the HEB condition in the oracle setting assuming that the $c$ parameter is given. We first give some notations. Denote by $L_i$ the smoothness constant of $f_i$, i.e., for all $x, y \in \Omega$ $f_i(x) - f_i(y) \leq \langle \nabla f_i(y), x - y \rangle + \frac{L_i}{2} \|x - y\|_2^2$. It implies that $f(x) \triangleq \frac{1}{n} \sum_{i=1}^n f_i(x)$ is also continuously differential convex function whose gradient is $L_f$-Lipschitz continuous, where $L_f \leq \frac{1}{n} \sum_{i=1}^n L_i$. For simplicity, we can take $L_f = \frac{1}{n} \sum_{i=1}^n L_i$. In the sequel, we let $L \triangleq \max_i L_i$ and assume that it is given or can be estimated for the problem. Denote by $\Omega_*$ the optimal set of the problem (1), and by $F_* = \min_{x \in \Omega} F(x)$. The detailed steps of SVRG under the HEB condition are presented in Algorithm 1. The formal guarantee of SVRG$^{\text{HEB}}$ is given in the following theorem.

**Theorem 2.** *Suppose problem (1) satisfies the HEB condition with $\theta \in (0, 1/2]$ and $F(x_0) - F_* \leq \epsilon_0$, where $x_0$ is an initial solution. Let $\eta = 1/(36L)$, and $T_1 \geq 81Lc^2 (1/\epsilon_0)^{1-2\theta}$. Algorithm 1 ensures*

$$\mathrm{E}[F(\bar{x}^{(R)}) - F_*] \leq (1/2)^R \epsilon_0. \tag{4}$$

*In particular, by running Algorithm 1 with $R = \lceil \log_2 \frac{\epsilon_0}{\epsilon} \rceil$, we have $\mathrm{E}[F(\bar{x}^{(R)}) - F_*] \leq \epsilon$, and the computational complexity for achieving an $\epsilon$-optimal solution in expectation is $O(n \log(\epsilon_0/\epsilon) + Lc^2 \max\{\frac{1}{\epsilon^{1-2\theta}}, \log(\epsilon_0/\epsilon)\})$.*

**Remark:** We make several remarks about the Algorithm 1 and the results in Theorem 2. First, the constant factors in $\eta$ and $T_1$ should not be treated literally, because we have made no effort to optimize them. Second, when $\theta = 1/2$ (i.e, the QEB condition holds), the Algorithm 1 reduces to the standard SVRG method under strong convexity, and the iteration complexity becomes $O((n + Lc^2) \log(\epsilon_0/\epsilon))$, which is the same as that of the standard SVRG with $Lc^2$ mimicking the condition number of the problem. Third, when $\theta = 0$ (i.e., with only the smoothness assumption), the Algorithm 1 reduces to SVRG++ [2] with one difference, where in SVRG$^{\text{HEB}}$ the initial point and the reference point for each outer loop are the same but are different in SVRG++, and the iteration complexity of SVRG$^{\text{HEB}}$ becomes $O(n \log(\epsilon_0/\epsilon) + \frac{Lc^2}{\epsilon})$ that is similar to that of SVRG++. Fourth, for intermediate

**Algorithm 1** SVRG method under HEB (SVRG$^{\text{HEB}}(x_0, T_1, R, \theta)$)

1: **Input**: $x_0 \in \Omega$, the number of inner initial iterations $T_1$, and the number of outer loops $R$.
2: $\bar{x}^{(0)} = x_0$
3: **for** $r = 1, 2, \ldots, R$ **do**
4:     $\bar{g}_r = \nabla f(\bar{x}^{(r-1)})$, $x_0^{(r)} = \bar{x}^{(r-1)}$
5:     **for** $t = 1, 2, \ldots, T_r$ **do**
6:       Choose $i_t \in \{1, \ldots, n\}$ uniformly at random.
7:       $g_t^{(r)} = \nabla f_{i_t}(x_{t-1}^{(r)}) - \nabla f_{i_t}(\bar{x}^{(r-1)}) + \bar{g}_r$.
8:       $x_t^{(r)} = \arg\min_{x \in \Omega} \langle g_t^{(r)}, x - x_{t-1}^{(r)} \rangle + \frac{1}{2\eta}\|x - x_{t-1}^{(r)}\|_2^2 + \Psi(x)$.
9:     **end for**
10:    $\bar{x}^{(r)} = \frac{1}{T_r} \sum_{t=1}^{T_r} x_t^{(r)}$
11:    $T_{r+1} = 2^{1-2\theta} T_r$
12: **end for**
13: **Output:** $\bar{x}^{(R)}$

---

**Algorithm 2** SVRG method under HEB with Restarting: SVRG$^{\text{HEB-RS}}$

1: **Input**: $x^{(0)} \in \Omega$, a small value $c_0 > 0$, and $\theta \in (0, 1/2)$.
2: **Initialization**: $T_1^{(1)} = 81Lc_0^2 (1/\epsilon_0)^{1-2\theta}$
3: **for** $s = 1, 2, \ldots, S$ **do**
4:    $x^{(s)}$=SVRG$^{\text{HEB}}$ $(x^{(s-1)}, T_1^{(s)}, R, \theta)$
5:    $T_1^{(s+1)} = 2^{1-2\theta} T_1^{(s)}$
6: **end for**

---

$\theta \in (0, 1/2)$ we can obtain faster convergence than SVRG++. Lastly, note that the number of iterations for each outer loop depends on the $c$ parameter in the HEB condition. The proof the Theorem 2 is simply built on previous analysis of SVRG and is deferred to the supplement.

## 4 Adaptive SVRG under the HEB condition in the dark setting

In this section, we will present adaptive variants of SVRG$^{\text{HEB}}$ that can be run in the dark setting, i.e, without assuming $c$ is known. We first present the variant for $\theta < 1/2$, which is simple and can help us understand the difficulty for $\theta = 1/2$.

### 4.1 Adaptive SVRG for $\theta \in (0, 1/2)$

An issue of SVRG$^{\text{HEB}}$ is that when $c$ is unknown the initial number of iterations $T_1$ in Algorithm 1 is difficult to estimate . A small value of $T_1$ may not guarantee SVRG$^{\text{HEB}}$ converges as Theorem 2 indicates. To address this issue, we can use the restarting trick, i.e, restarting SVRG$^{\text{HEB}}$ with an increasing sequences of $T_1$. The steps are shown in Algorithm 2. We can start with a small value of $c_0$, which is not necessarily larger than $c$. If $c_0$ is larger than $c$, the first call of SVRG$^{\text{HEB}}$ will yield an $\epsilon$-optimal solution as Theorem 2 indicates. Below, we assume that $c_0 \le c$.

**Theorem 3.** *Suppose problem (1) satisfies the HEB with $\theta \in (0, 1/2)$ and $F(x_0) - F_* \le \epsilon_0$, where $x_0$ is an initial solution. Let $c_0 \le c$, $\epsilon \le \frac{\epsilon_0}{2}$, $R = \lceil \log_2 \frac{\epsilon_0}{\epsilon} \rceil$ and $T_1^{(1)} = 81Lc_0^2 (1/\epsilon_0)^{1-2\theta}$. Then with at most a total number of $S = \left\lceil \frac{1}{\frac{1}{2}-\theta} \log_2 \left( \frac{c}{c_0} \right) \right\rceil + 1$ calls of SVRG$^{\text{HEB}}$ in Algorithm 2, we find a solution $x^{(S)}$ such that $\mathrm{E}[F(x^{(S)}) - F_*] \le \epsilon$. The computaional complexity of SVRG$^{\text{HEB-RS}}$ for obtaining such an $\epsilon$-optimal solution is $O\left( n \log(\epsilon_0/\epsilon) \log(c/c_0) + \frac{Lc^2}{\epsilon^{1-2\theta}} \right)$.*

**Remark:** The proof is in the supplement. We can see that Algorithm 2 cannot be applied to $\theta = 1/2$, which gives a constant sequence of $T_1^{(s)}$ and therefore cannot provide any convergence guarantee for a small value of $c_0 < c$. We have to develop a different variant for tackling $\theta = 1/2$. A minor point of worth mentioning is that if necessary we can stop Algorithm 2 appropriately by performing a proximal gradient update at $x^{(s)}$ (whose full gradient will be computed for the next stage) and checking if the proximal gradient's Euclidean norm square is less than a predefined level (c.f. (7)).

**Algorithm 3** SVRG method under QEB with Restarting and Search: SVRG$^{\text{QEB-RS}}$

1: **Input**: $\tilde{x}^{(0)} \in \Omega$, an initial value $c_0 > 0$, $\epsilon > 0$, $\rho = 1/\log(1/\epsilon)$ and $\vartheta \in (0,1)$.
2: $\bar{x}^{(0)} = \arg\min_{x \in \Omega} \langle \nabla f(\tilde{x}^0), x - \tilde{x}^0 \rangle + \frac{L}{2}\|x - \tilde{x}^0\|_2^2 + \Psi(x)$, $s = 0$
3: **while** $\|\bar{x}^{(s)} - \tilde{x}^{(s)}\|_2^2 > \epsilon$ **do**
4:     Set $R_s$ and $T_s = \lceil 81Lc_s^2 \rceil$ as in Lemma 2
5:     $\tilde{x}^{(s+1)}$=SVRG$^{\text{HEB}}(\bar{x}^{(s)}, T_s, R_s, 0.5)$
6:     $\bar{x}^{(s+1)} = \arg\min_{x \in \Omega} \langle \nabla f(\tilde{x}^{(s+1)}), x - \tilde{x}^{(s+1)} \rangle + \frac{L}{2}\|x - \tilde{x}^{(s+1)}\|_2^2 + \Psi(x)$
7:     $c_{s+1} = c_s$
8:     **if** $\|\bar{x}^{(s+1)} - \tilde{x}^{(s+1)}\|_2 \geq \vartheta\|\bar{x}^{(s)} - \tilde{x}^{(s)}\|_2$ **then**
9:         $c_{s+1} = \sqrt{2}c_s$, $\bar{x}^{(s+1)} = \bar{x}^{(s)}$, $\tilde{x}^{(s+1)} = \tilde{x}^{(s)}$
10:    **end if**
11:    $s = s + 1$
12: **end while**
13: **Output**: $\bar{x}^{(s)}$

## 4.2 Adaptive SVRG for $\theta = 1/2$

In light of the value of $T_1$ in Theorem 2 for $\theta = 1/2$, i.e., $T_1 = \lceil 81Lc^2 \rceil$, one might consider to start with a small value for $c$ and then increase its value by a constant factor at certain points in order to increase the value of $T_1$. But the challenge is to decide when we should increase the value of $c$. If one follows a similar procedure as in Algorithm 2, we may end up with a worse iteration complexity. To tackle this challenge, we need to develop an appropriate machinery to check whether the value of $c$ is already large enough for SVRG to decrease the objective value. However, we cannot afford the cost for computing the objective value due to large $n$. To this end, we develop a "certificate" that can be easily verified and can act as signal for a sufficient decrease in the objective value. The developed certificate is motivated by a property of proximal gradient update under the QEB as shown in (5).

**Lemma 1.** *Let $\bar{x} = \arg\min_{x \in \Omega} \langle \nabla f(\tilde{x}), x - \tilde{x} \rangle + \frac{L}{2}\|x - \tilde{x}\|_2^2 + \Psi(x)$. Then under the QEB condition of the problem (1), we have*

$$F(\bar{x}) - F_* \leq (L + L_f)^2 c^2 \|\bar{x} - \tilde{x}\|_2^2. \tag{5}$$

The above lemma indicates that we can perform a proximal gradient update at a point $\tilde{x}$ and use $\|\bar{x} - \tilde{x}\|_2$ as a gauge for monitoring the decrease in the objective value. However, the proximal gradient update is too expensive to compute due to the computation of full gradient $\nabla f(\tilde{x})$. Luckily, SVRG allows to compute the full gradient at a small number of reference points. We propose to leverage these full gradients to conduct the proximal gradient updates and develop the certificate for searching the value of $c$. The detailed steps of the proposed algorithm are presented in Algorithm 3 to which we refer as SVRG$^{\text{QEB-RS}}$. Similar to SVRG$^{\text{HEB-RS}}$, SVRG$^{\text{QEB-RS}}$ also calls SVRG$^{\text{HEB}}$ for multiple stages. We conduct the proximal gradient update at the returned solution of each SVRG$^{\text{HEB}}$, which also serves as the initial solution and the initial reference point for the next stage of SVRG$^{\text{HEB}}$ when our check in Step 7 fails. At each stage, at most $R_s + 1$ full gradients are computed, where $R_s$ is a logarithmic number as revealed later. Step 7 - Step 11 in Algorithm 3 are considered as our search step for searching the value of $c$. We will show that, if $c_s$ is larger than $c$, the condition in Step 7 is true with small probability. This can be seen from the following lemma.

**Lemma 2.** *Suppose problem (1) satisfies the QEB condition. Let $\mathcal{G}_0 \subseteq \mathcal{G}_1 \ldots \subseteq \mathcal{G}_s \ldots$ be a filtration with the sigma algebra $\mathcal{G}_s$ generated by all random events before line 4 of stage $s$ of Algorithm 3. Let $\eta = \frac{1}{36L}$, $T_s = \lceil 81Lc_s^2 \rceil$, $R_s = \left\lceil \log_2\left(\frac{2c_s^2(L+L_f)^2}{\vartheta^2 \rho L}\right) \right\rceil$. Then for any $\vartheta \in (0,1)$, we have*
$$\Pr\left(\|\bar{x}^{(s+1)} - \tilde{x}^{(s+1)}\|_2 \geq \vartheta\|\bar{x}^{(s)} - \tilde{x}^{(s)}\|_2 \Big| \mathcal{G}_s, c_s \geq c\right) \leq \rho.$$

*Proof.* By Lemma 1, we have $F(\bar{x}^{(s)}) - F_* \leq (L + L_f)^2 c^2 \|\bar{x}^{(s)} - \tilde{x}^{(s)}\|_2^2$ for all $s$. Below we consider stages such that $c_s \geq c$. Following Theorem 2 and the above inequality, when $T_s = \lceil 81Lc_s^2 \rceil \geq \lceil 81Lc^2 \rceil$, we have

$$\mathbb{E}[F(\tilde{x}^{(s+1)}) - F_*|\mathcal{G}_s] \leq 0.5^{R_s}(F(\bar{x}^{(s)}) - F_*) \leq 0.5^{R_s}(L + L_f)^2 c^2 \|\bar{x}^{(s)} - \tilde{x}^{(s)}\|_2^2. \tag{6}$$

Moreover, the smoothness of $f(x)$ and the definition of $\bar{x}^{(s+1)}$ imply (see Lemma 4 in the supplemnt).

$$F(\tilde{x}^{(s+1)}) - F_* \geq \frac{L}{2}\|\bar{x}^{(s+1)} - \tilde{x}^{(s+1)}\|_2^2. \tag{7}$$

By combining (7) and (6) and using Markov inequality, we have

$$\Pr\left(\frac{L}{2}\|\bar{x}^{(s+1)} - \tilde{x}^{(s+1)}\|_2^2 \geq \epsilon | \mathcal{G}_s\right) \leq \frac{0.5^{R_s}\,(L+L_f)^2\,c^2\|\bar{x}^{(s)} - \tilde{x}^{(s)}\|_2^2}{\epsilon}.$$

If we choose $\epsilon = \frac{\vartheta^2 L \|\bar{x}^{(s)} - \tilde{x}^{(s)}\|^2}{2}$ in the inequality above and let $R_s$ defined as in the assumption, the conclusion follows. $\qquad\square$

**Theorem 4.** *Under the same conditions as in Lemma 2 with $\rho = 1/\log(1/\epsilon)$, the expected computational complexity of SVRG$^{QEB\text{-}RS}$ for finding an $\epsilon$-optimal solution is at most*

$$O\left((Lc^2+n)\log_2\left(\frac{c^2(L+L_f)^2}{\vartheta^2 L}\log\left(\frac{1}{\epsilon}\right)\right)\left(\log_{1/\vartheta^2}\left(\frac{\|\bar{x}^{(0)} - \tilde{x}^{(0)}\|_2^2}{\epsilon}\right) + \log_2\left(\frac{c}{c_0}\right)\right)\right).$$

*Proof.* We call stage $s$ with $s = 0, 1, \ldots$ a *successful* stage if $\|\bar{x}^{(s+1)} - \tilde{x}^{(s+1)}\|_2 < \vartheta\|\bar{x}^{(s)} - \tilde{x}^{(s)}\|_2$; otherwise, the stage $s$ is called an *unsuccessful* stage. The condition $\|\bar{x}^{(s)} - \tilde{x}^{(s)}\|_2^2 \leq \epsilon$ will hold after $S_1 := \log_{1/\vartheta^2}\left(\frac{\|\bar{x}^{(0)} - \tilde{x}^{(0)}\|_2^2}{\epsilon}\right)$ successful stages and then Algorithm 3 will stop. Let $S$ denote the total number of stages when the algorithm stops. Although stage $s = S - 1$ is the last stage, for the convenience in the proof, we still define stage $s = S$ as a post-termination stage where no computation is performed.

In stage $s$ with $0 \leq s \leq S - 1$, the computational complexity is proportional to the number of stochastic gradient computations (#SGC), which is $T_s R_s + n(R_s + 1) \leq (T_s + 2n)R_s$. If stage $s$ is successful, then $R_{s+1} = R_s$ and $T_{s+1} = T_s$. If stage $s$ is unsuccessful, then $R_{s+1} = R_s + 1 \leq 2R_s$ and $T_{s+1} = 2T_s$ so that $R_{s+1}T_{s+1} \leq 4R_sT_s$. In either case, $R_s$ and $T_s$ are non-decreasing.

Note that, after $S_2 := \lceil 2\log_2(c/c_0)\rceil$ unsuccessful stages, we will have $c_s \geq c$. We will consider two scenarios: (I) the algorithm stops with $c_S < c$ and (I) the algorithm stops with $c_S \geq c$.

In the first scenario, we have $S_1$ successful stages and at most $S_2$ unsuccessfully stages so that $S \leq S_1 + S_2$ and $c_S < c$. The #SGC of all stages can be bounded by $(S_1 + S_2)(T_{S-1} + 2n)R_{S-1} \leq O\left(\left[\log_2(\frac{c}{c_0}) + \log_{1/\vartheta^2}\left(\frac{\|\bar{x}^{(0)} - \tilde{x}^{(0)}\|_2^2}{\epsilon}\right)\right]\log_2\left(\frac{2c^2(L+L_f)^2}{\vartheta^2\rho L}\right)(Lc^2+n)\right).$

Then, we consider the second scenario. Let $\hat{s}$ be the first stage with $c_s \geq c$, i.e., $\hat{s} := \min\{s|c_s \geq c\}$. It is easy to see that $c_{\hat{s}} < \sqrt{2}c$ and there are $S_2$ unsuccessful and less than $S_1$ successful stages before stage $\hat{s}$. Since the #SGC in any stage before $\hat{s}$ is bounded by $(T_{\hat{s}} + 2n)R_{\hat{s}} \leq O\left((Lc^2+n)\log_2\left(\frac{8c^2(L+L_f)^2}{\vartheta^2\rho L}\right)\right)$, the total #SGC in stages $0, 1, \ldots, \hat{s}-1$ is at most $(S_1+S_2)(T_{\hat{s}} + 2n)R_{\hat{s}} \leq O\left(\left[\log_2(\frac{c}{c_0}) + \log_{1/\vartheta^2}\left(\frac{\|\bar{x}^{(0)} - \tilde{x}^{(0)}\|_2^2}{\epsilon}\right)\right]\log_2\left(\frac{2c^2(L+L_f)^2}{\vartheta^2\rho L}\right)(Lc^2+n)\right).$

Next, we bound the total #SGC in stages $\hat{s}, \hat{s} + 1, \ldots, S$. In the rest of the proof, we consider stage $s$ with $\hat{s} \leq s \leq S$. We define $\mathcal{C}(\tilde{x}, \bar{x}, i, j, s)$ as the *expected* #SGC in stages $s, s+1, \ldots, S$, conditioning on that the initial state of stage $s$ are $\tilde{x}^{(s)} = \tilde{x}$ and $\bar{x}^{(s)} = \bar{x}$ and the numbers of successful and unsuccessful stages before stage $s$ are $i$ and $j$, respectively. Note that $s = i + j$. Because stage $s$ depends on the historical path only through the state variables $(\tilde{x}, \bar{x}, i, j, s)$, $\mathcal{C}(\tilde{x}, \bar{x}, i, j, s)$ is well defined and $(\tilde{x}, \bar{x}, i, j, s)$ transits in a Markov chain with the next state being $(\tilde{x}, \bar{x}, i, j + 1, s + 1)$ if stage $s$ does not succeed and being $(\tilde{x}_+, \bar{x}_+, i+1, j, s+1)$ if stage $s$ succeeds, where $\tilde{x}_+$=SVRG$^{HEB}(\bar{x}, T_s, R_s, 0.5)$ and $\bar{x}_+ = \arg\min_{x\in\Omega}\langle\nabla f(\tilde{x}_+), x - \tilde{x}_+\rangle + \frac{L}{2}\|x - \tilde{x}_+\|_2^2 + \Psi(x)$.

In the next, we will use backward induction to derive an upper bound for $\mathcal{C}(\tilde{x}, \bar{x}, i, j, s)$ that only depends on $i$ and $j$ but not on $s$, $\tilde{x}$ and $\bar{x}$. In particular, we want to show that

$$\mathcal{C}(\tilde{x}, \bar{x}, i, j, s) \leq \frac{4^{j-S_2}(T_{\hat{s}} + 2n)R_{\hat{s}}}{1 - 4\rho}A_i, \quad \text{for } i \geq 0,\ j \geq 0,\ i + j = s, s \geq \hat{s}, \qquad (8)$$

where $A_i := \sum_{r=0}^{S_1-i-1}\left(\frac{1-\rho}{1-4\rho}\right)^r$ if $0 \leq i \leq S_1 - 1$ and $A_i := 0$ if $i = S_1$.

We start with the base case where $i = S_1$. By definitions, the only stage with $i = S_1$ is the post-termination stage, namely, stage $s = S$. In this case, $\mathcal{C}(\tilde{x}, \bar{x}, i, j, s) = 0$ since stage $S$ performs no computation. Then, (8) holds trivially with $A_i = 0$.

Suppose $i < S_1$ and (8) holds for $i+1, i+2, \ldots, S_1$. We want to prove it also holds $i$. We define $X = X(\tilde{x}, \bar{x}, i, j, s)$ as the random variable that equals the number of unsuccessful stages from stage $s$ (including stage $s$) to the first successful stage among stages $s, s+1, s+2, \ldots, S-1$, conditioning on $s \geq \hat{s}$ and the state variables at the beginning of stage $s$ are $(\tilde{x}, \bar{x}, i, j, s)$. Note that $X = 0$ means stage $s$ is successful. For simplicity of notation, we use $\Pr(\cdot)$ to represent the conditional probability $\Pr(\cdot | s \geq \hat{s}, (\tilde{x}, \bar{x}, i, j, s))$. Since $c_s \geq c_{\hat{s}} \geq c$ for $s \geq \hat{s}$, we can show by Lemma 2 that [1]

$$
\begin{aligned}
\Pr(X = r) &= \left[\prod_{t=0}^{r-1} \Pr(X \geq t+1 | X \geq t)\right] \Pr(X = r | X \geq r), \\
\Pr(X \geq r+1 | X \geq r) &= \Pr(s+r \text{ fails } | \text{stages } s, s+1, \ldots, s+r-1 \text{ fail}) \leq \rho, \\
\Pr(X = r | X \geq r) &= \Pr(s+r \text{ succeeds } | \text{stages } s, s+1, \ldots, s+r-1 \text{ fail}), \\
&= 1 - \Pr(X \geq r+1 | X \geq r) \geq 1 - \rho.
\end{aligned}
\tag{9}
$$

When $X = r$, the #SGC from stage $s$ to the end of the algorithms will be $\sum_{t=0}^{r}(T_{s+t} + 2n)R_{s+t} + \mathbb{E}\mathcal{C}(\tilde{x}_+, \bar{x}_+, i+1, j+r, s+r+1)$, where $\mathbb{E}$ denotes the expectation over $\tilde{x}_+$ and $\bar{x}_+$ conditioning on $(\tilde{x}, \bar{x})$ and $\tilde{x}_+ = \text{SVRG}^{\text{HEB}}(\bar{x}, T_{s+r}, R_{s+r}, 0.5)$ and $\bar{x}_+ = \arg\min_{x \in \Omega} \langle \nabla f(\tilde{x}_+), x - \tilde{x}_+ \rangle + \frac{L}{2}\|x - \tilde{x}_+\|_2^2 + \Psi(x)$. Since stages $s, s+1, \ldots, s+r-1$ are unsuccessful, we have

$$
(T_{s+t} + 2n)R_{s+t} \leq 4^t (T_s + 2n)R_s \leq 4^{j+t-S_2}(T_{\hat{s}} + 2n)R_{\hat{s}} \text{ for } t = 0, 1, \ldots, r-1.
$$

Because (8) holds for $i+1$ and for any $\tilde{x}_+$ and $\bar{x}_+$, we have

$$
\mathcal{C}(\tilde{x}_+, \bar{x}_+, i+1, j+r, s+r+1) \leq \frac{4^{j+r-S_2}(T_{\hat{s}} + 2n)R_{\hat{s}}}{1 - 4\rho}A_{i+1}.
\tag{10}
$$

Based on the above inequality and the connection between $\mathcal{C}(\tilde{x}, \bar{x}, i, j, s)$ and $\mathcal{C}(\tilde{x}_+, \bar{x}_+, i+1, j+r, s+r+1)$, we will prove that (8) holds for $i, j, s$.

$$
\begin{aligned}
\mathcal{C}(\tilde{x}, \bar{x}, i, j, s) &= \sum_{r=0}^{\infty} \Pr(X = r)\left(\sum_{t=0}^{r}(T_{s+t} + 2n)R_{s+t} + \mathbb{E}\mathcal{C}(\tilde{x}_+, \bar{x}_+, i+1, j+r, s+r+1)\right) \\
&\leq \sum_{r=0}^{\infty} \Pr(X = r)\left(\sum_{t=0}^{r}(T_{s+t} + 2n)R_{s+t} + \frac{4^{j+r-S_2}(T_{\hat{s}} + 2n)R_{\hat{s}}}{1 - 4\rho}\frac{\left([(1-\rho)/(1-4\rho)]^{S_1-i-1} - 1\right)}{((1-\rho)/(1-4\rho) - 1)}\right) \\
&\leq \sum_{r=0}^{\infty} \Pr(X = r)\left(\sum_{t=0}^{r}4^{j+t-S_2}(T_{\hat{s}} + 2n)R_{\hat{s}} + \frac{4^{j+r-S_2}(T_{\hat{s}} + 2n)R_{\hat{s}}}{1 - 4\rho}A_{i+1}\right) \\
&\leq 4^{j-S_2}(T_{\hat{s}} + 2n)R_{\hat{s}}\sum_{r=0}^{\infty} \Pr(X = r)\left(\sum_{t=0}^{r}4^t + \frac{4^r}{1-4\rho}A_{i+1}\right) \\
&= 4^{j-S_2}(T_{\hat{s}} + 2n)R_{\hat{s}}\sum_{r=0}^{\infty}\left[\prod_{t=0}^{r-1} \Pr(X \geq t+1 | X \geq t)\right]\Pr(X = r | X \geq r)\left(\frac{4^{r+1} - 1}{3} + \frac{4^r A_{i+1}}{1 - 4\rho}\right).
\end{aligned}
$$

Since $1 - \rho \geq \frac{1}{4}$, for any $a \geq 0$ and any $b \geq a+1$, we have

$$
\begin{aligned}
&\left(\frac{4^{a+1} - 1}{3} + \frac{4^a A_{i+1}}{1 - 4\rho}\right) \\
&\leq (1 - \rho)\left(\frac{4^{a+2} - 1}{3} + \frac{4^{a+1} A_{i+1}}{1 - 4\rho}\right) \leq \Pr(X = a+1 | X \geq a+1)\left(\frac{4^{a+2} - 1}{3} + \frac{4^{a+1} A_{i+1}}{1 - 4\rho}\right) \\
&\leq \sum_{r=a+1}^{b}\left[\prod_{t=a+1}^{r-1} \Pr(X \geq t+1 | X \geq t)\right]\Pr(X = r | X \geq r)\left(\frac{4^{r+1} - 1}{3} + \frac{4^r A_{i+1}}{1 - 4\rho}\right) := \mathcal{D}_a^b,
\end{aligned}
$$

which implies

$$
\begin{aligned}
\mathcal{D}_{a-1}^b &:= \sum_{r=a}^{b}\left[\prod_{t=a}^{r-1} \Pr(X \geq t+1 | X \geq t)\right]\Pr(X = r | X \geq r)\left(\frac{4^{r+1} - 1}{3} + \frac{4^r A_{i+1}}{1 - 4\rho}\right) \\
&= \Pr(X = a | X \geq a)\left(\frac{4^{a+1} - 1}{3} + \frac{4^a A_{i+1}}{1 - 4\rho}\right) + \Pr(X \geq a+1 | X \geq a)\mathcal{D}_a^b \\
&\leq (1 - \rho)\left(\frac{4^{a+1} - 1}{3} + \frac{4^a A_{i+1}}{1 - 4\rho}\right) + \rho\mathcal{D}_a^b.
\end{aligned}
$$

Applying this inequality for $a = 0, 1, \ldots, b-1$ and the fact $\mathcal{D}^b_{b-1} \leq \frac{4^{b+1}-1}{3} + \frac{4^b A_{i+1}}{1-4\rho}$ gives

$$\mathcal{D}^b_{-1} \leq (1-\rho)\sum_{r=0}^{b-1} \rho^r \left( \frac{4^{r+1}-1}{3} + \frac{4^r A_{i+1}}{1-4\rho} \right) + \rho^b \left( \frac{4^{b+1}-1}{3} + \frac{4^b A_{i+1}}{1-4\rho} \right).$$

Since $4\rho < 1$, letting $b$ in the inequality above increase to infinity gives

$$\mathcal{C}(\tilde{x}, \bar{x}, i, j, s) \leq 4^{j-S_2}(T_{\hat{s}} + 2n)R_{\hat{s}}(1-\rho)\sum_{r=0}^{\infty} \rho^r \left( \frac{4^{r+1}-1}{3} + \frac{4^r A_{i+1}}{1-4\rho} \right)$$

$$= 4^{j-S_2}(T_{\hat{s}} + 2n)R_{\hat{s}} \left( \frac{1}{1-4\rho} + \frac{A_{i+1}(1-\rho)}{(1-4\rho)^2} \right) \frac{4^{j-S_2}(T_{\hat{s}} + 2n)R_{\hat{s}}A_i}{1-4\rho},$$

which is (8). Then by induction, (8) holds for any state $(\tilde{x}, \bar{x}, i, j, s)$ with $s \geq \hat{s}$. At the moment when the algorithm enters stage $\hat{s}$, we must have $j = S_2$ and $i = \hat{s} - S_2$. By (8) and the facts that $\hat{s} \geq S_2$ and that $A_i = \sum_{r=0}^{S_1-i-1} \left( \frac{1-\rho}{1-4\rho} \right)^r \leq (S_1 + S_2 - \hat{s}) \left( \frac{1-\rho}{1-4\rho} \right)^{S_1+S_2-\hat{s}}$, the expected #SGC from stage $\hat{s}$ to the end of algorithm is

$$\mathcal{C}(\tilde{x}, \bar{x}, \hat{s} - S_2, S_2, \hat{s}) \leq \frac{(T_{\hat{s}} + 2n)R_{\hat{s}}}{1-4\rho}(S_1 + S_2 - \hat{s}) \left( \frac{1-\rho}{1-4\rho} \right)^{S_1+S_2-\hat{s}}$$

$$\leq O\left( (Lc^2 + n)\log_2\left( \frac{8c^2(L+L_f)^2}{\vartheta^2 \rho L} \right) S_1 \left( \frac{1-\rho}{1-4\rho} \right)^{S_1} \right).$$

In light of the value of $\rho$, i.e., $\rho = \frac{1}{\log(1/\epsilon)}$, we have $\left( \frac{1-\rho}{1-4\rho} \right)^{S_1} = \left( \frac{\|\bar{x}^{(0)} - \tilde{x}^{(0)}\|_2^2}{\epsilon} \right)^{\frac{\log\left( \frac{1-\rho}{1-4\rho} \right)}{\log 1/\vartheta^2}} \leq \left( \frac{\|\bar{x}^{(0)} - \tilde{x}^{(0)}\|_2}{\epsilon} \right)^{\frac{3\rho}{(1-4\rho)\log 1/\vartheta}} = O\left( \left( \frac{1}{\epsilon} \right)^{3\rho} \right) \leq O(1)$. Therefore, by adding the #SGC before and after the $\hat{s}$ stages in the second scenario, we have the expected total #SGC is $O\left( \left( \log\left( \frac{c}{c_0} \right) + \log\left( \frac{\|\bar{x}^{(0)} - \tilde{x}^{(0)}\|_2^2}{\epsilon} \right) \right) \log\left( \frac{c^2(L+L_f)^2}{\rho L} \right)(Lc^2 + n) \right)$. □

## 5  Applications and Experiments

In this section, we consider some applications in machine learning and present some experimental results. We will consider finite-sum problems in machine learning where $f_i(x) = \ell(x^\top a_i, b_i)$ denotes a loss function on an observed training feature and label pair $(a_i, b_i)$, and $\Psi(x)$ denotes a regularization on the model $x$. Let us first consider some examples of loss functions and regularizers that satisfy the QEB condition. More examples can be found in [29, 28, 27, 14].

**Piecewise convex quadratic (PCQ) problems.** According to the global error bound of piecewise convex polynomials by Li [10], PCQ problems satisfy the QEB condition. Examples of such problems include empirical square loss, squared hinge loss or Huber loss minimization with $\ell_1$ norm, $\ell_\infty$ norm or $\ell_{1,\infty}$ norm regularization or constraint.

**A family of structured smooth composite functions.** This family include functions of the form $F(x) = h(Ax) + \Psi(x)$, where $\Psi(x)$ is a polyhedral function or an indicator function of a polyhedral set and $h(\cdot)$ is a smooth and strongly convex function on any compact set. Accoding to studies in [6, 20], the QEB holds on any compact set or the involved polyhedral set. Examples of interesting loss functions include the aforementioned square loss and the logisitc loss as well.

For examples satisfying the HEB condition with intermediate values of $\theta \in (0, 1/2)$, we can consider $\ell_1$ constrained $\ell_p$ norm regression, where the objective $f(x) = 1/n \sum_{i=1}^n (x^\top a_i - b_i)^p$ with $p \in 2\mathbb{N}^+$ [23]. According to the reasoning in [14], the HEB condition holds with $\theta = 1/p$.

Before presenting the experimental results, we would like to remark that in many regularized machine learning formulations, no constraint in a compact domain $x \in \Omega$ is included. Nevertheless, we can explicitly add a constraint $\Psi(x) \leq B$ into the problem to ensure that intermediate solutions generated by the proposed algorithms always stay in a compact set, where $B$ can be set to a large value without affecting the optimal solutions. The proximal mapping of $\Psi(x)$ with such an explicit constraint can be efficiently handled by combining the proximal mapping and a binary search for the Lagrangian

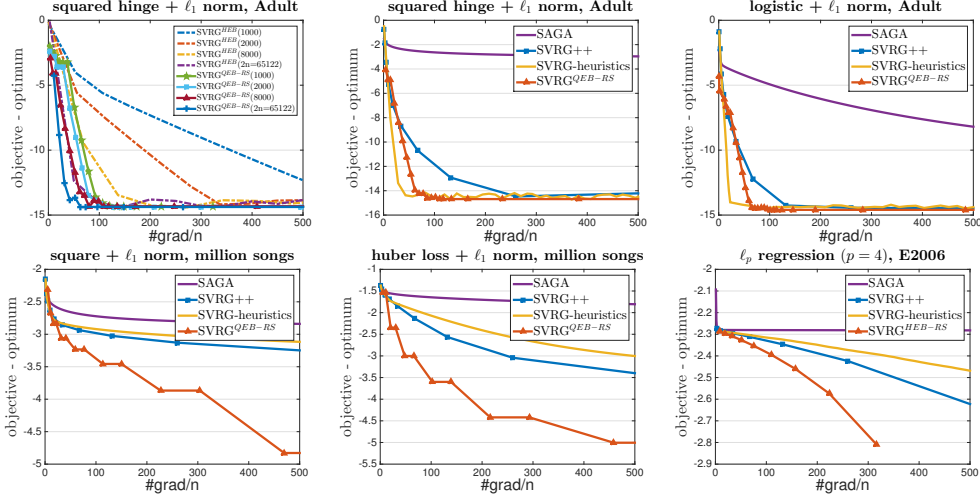

Figure 1: Comparison of different algorithms for solving different problems on different datasets.

multiplier. In practice, as long as $B$ is sufficiently large, the constraint remains inactive and the computational cost remains the same.

Next, we conduct some experiments to demostrate the effectiveness of the proposed algorithms on several tasks, including $\ell_1$ regularized squared hinge loss minimization, $\ell_1$ regularized logistic loss minimization for linear classification problems; and $\ell_1$ constrained $\ell_p$ norm regression, $\ell_1$ regularized square loss minimization and $\ell_1$ regularized Huber loss minimization for linear regression problems. We use three datasets from libsvm website: Adult ($n = 32561, d = 123$), E2006-tfidf ($n = 16087, d = 150360$), and YearPredictionMSD ($n = 51630, d = 90$). Note that we use the testing set of YearPredictionMSD data for our experiment because some baselines need a lot of time to converge on the large training set. We set the regularization parameter of $\ell_1$ norm and the upper bound of $\ell_1$ constraint to be $10^{-4}$ and 100, respectively. In each plot, the difference between objective value and optimum is presented in log scale.

Our first experiment is to justify the proposed SVRG$^{\text{QEB-RS}}$ algorithm by comparing it with SVRG$^{\text{HEB}}$ with different estimations of $c$ (corresponding to the different initial values of $T_1$). We try four different values of $T_1 \in \{1000, 2000, 8000, 2n\}$. The result is plotted in the top left of Figure 1. We can see that SVRG$^{\text{HEB}}$ with some underestimated values of $T_1$ (e.g, 1000, 2000) converge very slowly. However, the performance of SVRG$^{\text{QEB-RS}}$ is not affected too much by the initial value of $T_1$, which is consistent with our theory showing the log dependence on the initial value of $c$. Moreover, SVRG$^{\text{QEB-RS}}$ with different values of $T_1$ perform always better than their counterparts of SVRG$^{\text{HEB}}$.

Then we compare SVRG$^{\text{QEB-RS}}$ and SVRG$^{\text{HEB-RS}}$ to other baselines for solving different problems on different data sets. We choose SAGA, SVRG++ as the baselines. We also notice that a heuristic variant of SVRG++ was suggested in [2] where epoch length is automatically determined based on the change in the variance of gradient estimators between two consecutive epochs. However, according to our experiments we find that this heuristic automatic strategy cannot always terminate one epoch because their suggested criterion cannot be met. This is also confirmed by our communication with the authors of SVRG++. To make it work, we manually add an upper bound constraint of each epoch length equal to $2n$ following the suggestion in [8]. The resulting baseline is denoted by SVRG-heuristics. For all algorithms, the step size is best tuned. The initial epoch length of SVRG++ is set to $n/4$ following the suggestion in [2], and the same initial epoch length is also used in our algorithms. The comparison with these baselines are reported in remaining figures of Figure 1. We can see that SVRG$^{\text{QEB-RS}}$ (resp. SVRG$^{\text{HEB-RS}}$) always has superior performance, while SVRG-heuristics sometimes performs well sometimes bad.

## Acknowlegements

We thank the anonymous reviewers for their helpful comments. Y. Xu and T. Yang are partially supported by National Science Foundation (IIS-1463988, IIS-1545995).

## Footnotes

[1] We follow the convention that $\prod_i^j = 1$ if $j < i$.

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
