[Supplementary Material · AdaSVRG-supplement.pdf]

# Supplementary Materials
# "Adaptive SVRG Methods under Error Bound Conditions with Unknown Growth Parameter"

**Yi Xu[†], Qihang Lin[‡], Tianbao Yang[†]**
[†]Department of Computer Science, The University of Iowa, Iowa City, IA 52242, USA
[‡]Department of Management Sciences, The University of Iowa, Iowa City, IA 52242, USA
{yi-xu, qihang-lin, tianbao-yang}@uiowa.edu

## 1  Proof of Theorem 2

**Theorem 2.** *Assume that the problem (1) satisfies the HEB condition with $\theta \in (0, 1/2]$ and $F(x_0) - F_* \leq \epsilon_0$, where $x_0$ is an initial solution. Let $\eta = 1/(36L)$, and $T_1 \geq 81Lc^2 \left(1/\epsilon_0\right)^{1-2\theta}$. Then Algorithm 1 ensures*

$$\mathrm{E}[F(\bar{x}^{(R)}) - F_*] \leq (1/2)^R \epsilon_0. \tag{11}$$

*In particular, by running Algorithm 1 with $R = \lceil \log_2 \frac{\epsilon_0}{\epsilon} \rceil$, we have $\mathrm{E}[F(\bar{x}^{(R)}) - F_*] \leq \epsilon$, and the computational complexity for achieving an $\epsilon$-optimal solution in expectation is $O(n \log(\epsilon_0/\epsilon) + Lc^2 \max\{\frac{1}{\epsilon^{1-2\theta}}, \log(\epsilon_0/\epsilon)\})$.*

We need the following lemma to prove Theorem 2, which has been established in previous work [2].

**Lemma 3.** *For the $r$-th outer loop of Algorithm 1, for any $x_* \in \Omega_*$ we have*

$$2\eta(1 - 4L\eta)T_r \mathrm{E}[F(\bar{x}^{(r)}) - F(x_*)] \leq \mathrm{E}[\|\bar{x}^{(r-1)} - x_*\|_2^2] + 8L\eta^2(T_r + 1)\mathrm{E}[F(\bar{x}^{(r-1)}) - F(x_*)]. \tag{12}$$

*Proof of Theorem 2.* Denote by $\epsilon_r = \epsilon_0/2^r$. We will prove (11) by induction. Assume that $\mathrm{E}[F(\bar{x}^{(r-1)}) - F(x_*)] \leq \epsilon_{r-1}$, which is true for $r = 1$. Let $x_*$ in Lemma 3 be the closest optimal solution to $\bar{x}^{(r-1)}$. Taking expectation over all random variables on both sides of (12), we get

$$\mathrm{E}[F(\bar{x}^{(r)}) - F_*] \leq \frac{1}{2\eta(1 - 4L\eta)T_r}\mathrm{E}\|\bar{x}^{(r-1)} - x_*\|_2^2 + \frac{4L\eta(T_r + 1)}{(1 - 4L\eta)T_r}\mathrm{E}[F(\bar{x}^{(r-1)}) - F_*]$$

$$\leq \frac{1}{2\eta(1 - 4L\eta)T_r}c^2 \mathrm{E}[F(\bar{x}^{(r-1)}) - F_*]^{2\theta} + \frac{4L\eta(T_r + 1)}{(1 - 4L\eta)T_r}\mathrm{E}[F(\bar{x}^{(r-1)}) - F_*]$$

$$\leq \frac{1}{2\eta(1 - 4L\eta)T_r}c^2 (\mathrm{E}[F(\bar{x}^{(r-1)}) - F_*])^{2\theta} + \frac{4L\eta(T_r + 1)}{(1 - 4L\eta)T_r}\mathrm{E}[F(\bar{x}^{(r-1)}) - F_*],$$

where the second inequality uses the HEB condition and the last inequality uses the concavity of $x^{2\theta}$ for $x \geq 0$ and $2\theta \leq 1$. By noting the values of $\eta = \frac{1}{36L}$ and $T_r \geq 81Lc^2\epsilon_{r-1}^{2\theta-1}$,

$$\frac{1}{2\eta(1 - 4L\eta)T_r}c^2\epsilon_{r-1}^{2\theta} \leq \frac{\epsilon_{r-1}}{4}, \quad \frac{4L\eta(T_r + 1)}{(1 - 4L\eta)T_r}\epsilon_{r-1} \leq \frac{\epsilon_{r-1}}{4}.$$

Thus $\mathrm{E}[F(\bar{x}^{(r)}) - F_*] \leq \frac{\epsilon_{r-1}}{2} \triangleq \epsilon_r$. We can complete the proof in light of $R = \lceil \log_2 \frac{\epsilon_0}{\epsilon} \rceil$.   □

## 2 Proof of Lemma 3

*Proof.* First, we can write the update of $x_t^{(r)} = \arg\min_{x \in \mathbb{R}^d} \frac{1}{2}\|x - (x_{t-1}^{(r)} - \eta g_t^{(r)})\|_2^2 + \eta\Psi(x)$, and we know that $\frac{1}{2}\|x - (x_{t-1}^{(r)} - \eta g_t^{(r)})\|_2^2 + \eta\Psi(x)$ is 1-strongly convex w.r.t. $\|\cdot\|_2$ in terms of $x$. By the first-order optimilaty condition, for any $x$ we get

$$\frac{1}{2}\|x - (x_{t-1}^{(r)} - \eta g_t^{(r)})\|_2^2 + \eta\Psi(x) \geq \frac{1}{2}\|x_t^{(r)} - (x_{t-1}^{(r)} - \eta g_t^{(r)})\|_2^2 + \eta\Psi(x_t^{(r)}) + \frac{1}{2}\|x_t^{(r)} - x\|_2^2.$$

Rewrite above inequality, then

$$
\begin{aligned}
\eta\Psi(x_t^{(r)}) - \eta\Psi(x) \leq & \frac{1}{2}\|x_{t-1}^{(r)} - x\|_2^2 - \frac{1}{2}\|x_t^{(r)} - x\|_2^2 - \frac{1}{2}\|x_t^{(r)} - x_{t-1}^{(r)}\|_2^2 + \eta\langle g_t^{(r)}, x - x_t^{(r)}\rangle \\
= & \frac{1}{2}\|x_{t-1}^{(r)} - x\|_2^2 - \frac{1}{2}\|x_t^{(r)} - x\|_2^2 + \eta\langle g_t^{(r)} - \nabla f(x_{t-1}^{(r)}), x - x_t^{(r)}\rangle \\
& + \eta\langle \nabla f(x_{t-1}^{(r)}), x_{t-1}^{(r)} - x_t^{(r)}\rangle - \frac{1}{2}\|x_t^{(r)} - x_{t-1}^{(r)}\|_2^2 \\
& + \eta\langle \nabla f(x_{t-1}^{(r)}), x - x_{t-1}^{(r)}\rangle.
\end{aligned}
\tag{13}
$$

Since $f$ is $L$-smooth and $0 < \eta \leq \frac{1}{L}$,

$$
\begin{aligned}
f(x_t^{(r)}) - f(x_{t-1}^{(r)}) \leq & \langle \nabla f(x_{t-1}^{(r)}), x_t^{(r)} - x_{t-1}^{(r)}\rangle + \frac{L}{2}\|x_t^{(r)} - x_{t-1}^{(r)}\|_2^2 \\
\leq & \langle \nabla f(x_{t-1}^{(r)}), x_t^{(r)} - x_{t-1}^{(r)}\rangle + \frac{1}{2\eta}\|x_t^{(r)} - x_{t-1}^{(r)}\|_2^2.
\end{aligned}
$$

That is,

$$\eta\langle \nabla f(x_{t-1}^{(r)}), x_{t-1}^{(r)} - x_t^{(r)}\rangle - \frac{1}{2}\|x_t^{(r)} - x_{t-1}^{(r)}\|_2^2 \leq \eta[f(x_{t-1}^{(r)}) - f(x_t^{(r)})]. \tag{14}$$

By the convexity of $f$, we get

$$\langle \nabla f(x_{t-1}^{(r)}), x - x_{t-1}^{(r)}\rangle \leq f(x) - f(x_{t-1}^{(r)}). \tag{15}$$

Plugging in inequalities (14) and (15) into inequality (13), we get

$$
\begin{aligned}
F(x_t^{(r)}) - F(x) \leq & \frac{1}{2\eta}\|x_{t-1}^{(r)} - x\|_2^2 - \frac{1}{2\eta}\|x_t^{(r)} - x\|_2^2 - \langle g_t^{(r)} - \nabla f(x_{t-1}^{(r)}), x_t^{(r)} - x\rangle \\
= & \frac{1}{2\eta}\|x_{t-1}^{(r)} - x\|_2^2 - \frac{1}{2\eta}\|x_t^{(r)} - x\|_2^2 - \langle g_t^{(r)} - \nabla f(x_{t-1}^{(r)}), \widehat{x}_t^{(r)} - x\rangle \\
& - \langle g_t^{(r)} - \nabla f(x_{t-1}^{(r)}), x_t^{(r)} - \widehat{x}_t^{(r)}\rangle \\
\leq & \frac{1}{2\eta}\|x_{t-1}^{(r)} - x\|_2^2 - \frac{1}{2\eta}\|x_t^{(r)} - x\|_2^2 - \langle g_t^{(r)} - \nabla f(x_{t-1}^{(r)}), \widehat{x}_t^{(r)} - x\rangle \\
& + \|g_t^{(r)} - \nabla f(x_{t-1}^{(r)})\|_2 \|x_t^{(r)} - \widehat{x}_t^{(r)}\|_2 \\
\leq & \frac{1}{2\eta}\|x_{t-1}^{(r)} - x\|_2^2 - \frac{1}{2\eta}\|x_t^{(r)} - x\|_2^2 - \langle g_t^{(r)} - \nabla f(x_{t-1}^{(r)}), \widehat{x}_t^{(r)} - x\rangle \\
& + \|g_t^{(r)} - \nabla f(x_{t-1}^{(r)})\|_2 \|x_{t-1}^{(r)} - \eta g_t^{(r)} - (x_{t-1}^{(r)} - \eta\nabla f(x_{t-1}^{(r)}))\|_2 \\
= & \frac{1}{2\eta}\|x_{t-1}^{(r)} - x\|_2^2 - \frac{1}{2\eta}\|x_t^{(r)} - x\|_2^2 - \langle g_t^{(r)} - \nabla f(x_{t-1}^{(r)}), \widehat{x}_t^{(r)} - x\rangle \\
& + \eta\|g_t^{(r)} - \nabla f(x_{t-1}^{(r)})\|_2^2,
\end{aligned}
\tag{16}
$$

where $\widehat{x}_t^{(r)} = \arg\min_{x \in \mathbb{R}^d} \frac{1}{2}\|x - (x_{t-1}^{(r)} - \eta\nabla f(x_{t-1}^{(r)}))\|_2^2 + \eta\Psi(x)$. Please notice that the update of $\widehat{x}_t^{(r)}$ is not used in the Algorithm, but only for analysis. Letting $x = x_*$ and taking expectation over both sides, we have

$$
\begin{aligned}
2\eta\mathrm{E}[F(x_t^{(r)}) - F(x_*)] \leq & \|x_{t-1}^{(r)} - x_*\|_2^2 - \mathrm{E}[\|x_t^{(r)} - x_*\|_2^2] + 2\eta^2\mathrm{E}[\|g_t^{(r)} - \nabla f(x_{t-1}^{(r)})\|_2^2] \\
\leq & \|x_{t-1}^{(r)} - x_*\|_2^2 - \mathrm{E}[\|x_t^{(r)} - x_*\|_2^2] \\
& + 8L\eta^2[F(x_{t-1}^{(r)}) - F(x_*) + F(\bar{x}^{(r-1)}) - F(x_*)],
\end{aligned}
$$

where we use the fact that $\mathrm{E}[\langle g_t^{(r)} - \nabla f(x_{t-1}^{(r)}), \widehat{x}_t^{(r)} - x \rangle] = 0$ and use Corollary 3.5 in [2] to upper bound the expected variance $\mathrm{E}[\|g_t^{(r)} - \nabla f(x_{t-1}^{(r)})\|_2^2]$. Then

$$
\begin{aligned}
\mathrm{E}[\|x_t^{(r)} - x_*\|_2^2] \leq & \|x_{t-1}^{(r)} - x_*\|_2^2 - 2\eta \mathrm{E}[F(x_t^{(r)}) - F(x_*)] \\
& + 8L\eta^2[F(x_{t-1}^{(r)}) - F(x_*) + F(\bar{x}^{(r-1)}) - F(x_*)].
\end{aligned}
\tag{17}
$$

For a fixed $r$, by summing the previous inequality over $t = 1, \ldots, T$ and taking expectation with respect to the history of random variables sequence $i_1, i_2, \ldots, i_T$, we obtain

$$
\begin{aligned}
2\eta(1 - 4L\eta) & \sum_{t=1}^{T-1} \mathrm{E}[F(x_t^{(r)}) - F(x_*)] \\
\leq & \|x_0^{(r)} - x_*\|_2^2 - \mathrm{E}[\|x_T^{(r)} - x_*\|_2^2] - 2\eta \mathrm{E}[F(x_T^{(r)}) - F(x_*)] \\
& + 8L\eta^2[F(x_0^{(r)}) - F(x_*) + T(F(\bar{x}^{(r-1)}) - F(x_*))] \\
\leq & \|x_0^{(r)} - x_*\|_2^2 + 8L\eta^2[F(x_0^{(r)}) - F(x_*) + T(F(\bar{x}^{(r-1)}) - F(x_*))] \\
= & \|x_0^{(r)} - x_*\|_2^2 + 8L\eta^2(T+1)[F(x_0^{(r)}) - F(x_*)],
\end{aligned}
\tag{18}
$$

where the last inequality uses the facts that $-\mathrm{E}[\|x_T^{(r)} - x_*\|_2^2] \leq 0$ and $-2\eta \mathrm{E}[F(x_T^{(r)}) - F(x_*)] \leq 0$, and the last equality uses $x_0^{(r)} = \bar{x}^{(r-1)}$. By the convexity of $F(x)$ and the defination of $\bar{x}^{(r)}$ and $x_0^{(r)} = \bar{x}^{(r-1)}$ we have

$$
2\eta(1 - 4L\eta)T\mathrm{E}[F(\bar{x}^{(r)}) - F(x_*)] \leq \|\bar{x}^{(r-1)} - x_*\|_2^2 + 8L\eta^2(T+1)[F(\bar{x}^{(r-1)}) - F(x_*)].
\tag{19}
$$

$\square$

# 3 Proof of Theorem 3

**Theorem 3.** *Assume that the problem (1) satisfies the HEB with $\theta \in (0, 1/2)$ and $F(x_0) - F_* \leq \epsilon_0$, where $x_0$ is an initial solution, and $c_0 \leq c$. Let $\epsilon \leq \frac{\epsilon_0}{2}$, $R = \lceil \log_2 \frac{\epsilon_0}{\epsilon} \rceil$ and $T_1^{(1)} = 81Lc_0^2 (1/\epsilon_0)^{1-2\theta}$. Then with at most a total number of $S = \left\lceil \frac{1}{\frac{1}{2} - \theta} \log_2 \left( \frac{c}{c_0} \right) \right\rceil + 1$ calls of $SVRG^{HEB}$ in Algorithm 2, we find a solution $x^{(S)}$ such that $\mathrm{E}[F(x^{(S)}) - F_*] \leq \epsilon$. The computaional complexity of $SVRG^{HEB\text{-}RS}$ for obtaining such an $\epsilon$-optimal solution is $O\left( n \log(\epsilon_0/\epsilon) \log(c/c_0) + \frac{Lc^2}{\epsilon^{1-2\theta}} \right)$.*

*Proof.* Denote by $c_{s+1} = 2^{\frac{1-2\theta}{2}} c_s$. Since $c \geq c_0$ and $\frac{2}{1-2\theta} > 2$, we have $F(x_0) - F_* \leq \epsilon_0 \left( \frac{c}{c_0} \right)^{\frac{2}{1-2\theta}}$. Following the proof of Theorem 2, we can show that

$$
\mathrm{E}[F(x^{(1)}) - F_*] \leq \left( \frac{1}{2} \right)^R \epsilon_0 \left( \frac{c}{c_0} \right)^{\frac{2}{1-2\theta}} = \epsilon \left( \frac{c}{c_0} \right)^{\frac{2}{1-2\theta}}
\tag{20}
$$

with $R = \lceil \log_2 \frac{\epsilon_0}{\epsilon} \rceil$ and $T_1^{(1)} = 81Lc_0^2 \left( \frac{1}{\epsilon_0} \right)^{1-2\theta} = 81Lc^2 \left( \frac{1}{\epsilon_0 \left( \frac{c}{c_0} \right)^{\frac{2}{1-2\theta}}} \right)^{1-2\theta}$. Next, since $\epsilon \leq \frac{\epsilon_0}{2}$,

then we have $\mathrm{E}[F(x^{(1)}) - F_*] \leq \frac{\epsilon_0}{2} \left( \frac{c}{c_0} \right)^{\frac{2}{1-2\theta}} = \epsilon_0 \left( \frac{c}{c_1} \right)^{\frac{2}{1-2\theta}}$. By running $SVRG^{heb}$ from $x^{(1)}$ with $T_1^{(2)} = 81Lc_1^2 \left( \frac{1}{\epsilon_0} \right)^{1-2\theta} = 81Lc^2 \left( \frac{1}{\epsilon_0 \left( \frac{c}{c_1} \right)^{\frac{2}{1-2\theta}}} \right)^{1-2\theta}$, Theorem 2 ensures that

$$
\mathrm{E}[F(x^{(2)}) - F_*] \leq \left( \frac{1}{2} \right)^R \epsilon_0 \left( \frac{c}{c_1} \right)^{\frac{2}{1-2\theta}} = \epsilon \left( \frac{c}{c_1} \right)^{\frac{2}{1-2\theta}}.
\tag{21}
$$

By continuing the process, with $S = \left\lceil \frac{2}{1-2\theta} \log_2 \left( \frac{c}{c_0} \right) \right\rceil + 1$, we have

$$\mathrm{E}[F(x^{(S)}) - F_*] \leq \left( \frac{1}{2} \right)^R \epsilon_0 \left( \frac{c}{c_{S-1}} \right)^{\frac{2}{1-2\theta}} = \epsilon \left( \frac{c}{c_{S-1}} \right)^{\frac{2}{1-2\theta}} \leq \epsilon. \tag{22}$$

The total number of iterations for the $S$ calls of SVRG$^{\mathrm{heb}}$ is upper bounded by

$$
\begin{aligned}
T_{\mathrm{total}} &= \sum_{s=0}^{S-1} (nR + \sum_{r=1}^{R} T_1^{(s+1)} 2^{(1-2\theta)(r-1)}) = nRS + \sum_{s=0}^{S-1} T_1^{(s+1)} \sum_{r=1}^{R} 2^{(1-2\theta)(r-1)} \\
&= nRS + \sum_{s=0}^{S-1} T_1^{(1)} 2^{(1-2\theta)s} \sum_{r=1}^{R} 2^{(1-2\theta)(r-1)} \\
&\leq O \left( n\log(\epsilon_0/\epsilon) \log(c/c_0) + \left( \frac{c}{c_0} \right)^2 \left( \frac{\epsilon_0}{\epsilon} \right)^{1-2\theta} T_1^{(1)} \right) \\
&\leq O \left( n\log(\epsilon_0/\epsilon) \log(c_0) + \frac{Lc^2}{\epsilon^{1-2\theta}} \right).
\end{aligned}
$$

$\square$

## 4  Omitted Proof of Lemma 2

**Lemma 4.** *Let* $\bar{x} = \arg\min_{x\in\Omega} \langle \nabla f(\tilde{x}), x - \tilde{x} \rangle + \frac{L}{2} \|x - \tilde{x}\|_2^2 + \Psi(x)$. *Assume that* $f(x)$ *is* $L$-*smooth, we have*

$$F(\tilde{x}) - F_* \geq \frac{L}{2} \|\bar{x} - \tilde{x}\|^2. \tag{23}$$

*Proof.* Since $f(x)$ is $L$-smooth, then we get

$$f(\bar{x}) - f(\tilde{x}) \leq \langle \nabla f(\tilde{x}), \bar{x} - \tilde{x} \rangle + \frac{L}{2} \|\bar{x} - \tilde{x}\|_2^2. \tag{24}$$

By the defination of $\bar{x}$ and the strong convexity of $L(x) = \langle \nabla f(\tilde{x}), x - \tilde{x} \rangle + \frac{L}{2} \|x - \tilde{x}\|_2^2 + \Psi(x)$, we have

$$\langle \nabla f(\tilde{x}), \bar{x} - \tilde{x} \rangle + \frac{L}{2} \|\bar{x} - \tilde{x}\|_2^2 + \Psi(\bar{x}) \leq \Psi(\tilde{x}) - \frac{L}{2} \|\bar{x} - \tilde{x}\|_2^2. \tag{25}$$

Combining inequalities (24) and (25) with the fact that $F(x) = f(x) + \Psi(x)$ yeilds

$$F(\tilde{x}) - F(\bar{x}) \geq \frac{L}{2} \|\bar{x} - \tilde{x}\|^2.$$

We complete the proof by using $F(\bar{x}) \geq F_*$. $\square$

## 5  Proof of Lemma 1

**Lemma 1.** *Let* $\bar{x} = \arg\min_{x\in\Omega} \langle \nabla f(\tilde{x}), x - \tilde{x} \rangle + \frac{L}{2} \|x - \tilde{x}\|_2^2 + \Psi(x)$. *Then under the QEB condition of the problem (1), we have*

$$F(\bar{x}) - F_* \leq (L + L_f)^2 c^2 \|\bar{x} - \tilde{x}\|_2^2. \tag{26}$$

Before delving into the detailed analysis, we first present some lemmas.

**Lemma 5** (Theorem 1 [1]). *For a constant* $L > 0$ *and* $y \in \Omega$, *if*

$$v = \arg\min_{z\in\Omega} \left\{ f(y) + \langle \nabla f(y), z - y \rangle + \frac{L}{2} \|z - y\|_2^2 + \Psi(z) \right\},$$

*then for any* $x \in \Omega$,

$$\langle F'(v), x - v \rangle \geq -(L + L_f) \|v - y\|_2 \|v - x\|_2. \tag{27}$$

*Proof.* By the first order optimality condition, for any $x \in \Omega$,

$$\langle \nabla f(y) + \Psi^{'}(v) + L(v - y), x - v \rangle \geq 0,$$

where $\Psi^{'}(v) \in \partial \Psi(v)$, the set of subgradient of $\Psi$ at $v$. Then

$$\begin{aligned}
\langle \nabla f(v) + \Psi^{'}(v), v - x \rangle &\leq \langle \nabla f(v) - \nabla f(y) - L(v - y), v - x \rangle \\
&= \langle \nabla f(v) - \nabla f(y), v - x \rangle - L\langle v - y, v - x \rangle \\
&\leq \|\nabla f(v) - \nabla f(y)\|_2 \|v - x\|_2 + L\|v - y\|_2 \|v - x\|_2 \\
&\leq (L_f + L)\|v - y\|_2 \|v - x\|_2.
\end{aligned}$$

where the last inequality uses the smoothness of $f$. We complete the proof by using $F'(v) = \nabla f(v) + \Psi^{'}(v)$. □

**Lemma 6.** *Suppose that the problem (1) satisfies the QEB condition (2) and then for any $y$, $v$ defined in Lemma 5, we have*

$$\|v - v_*\|_2 \leq (L_f + L)c^2 \|v - y\|_2, \tag{28}$$

*where $v_*$ is the closest optimal solution to $v$.*

*Proof.* By the proof of Lemma 5, we have

$$\begin{aligned}
(L_f + L)\|v - y\|_2 \|v - v_*\|_2 &\geq \langle \nabla f(v) + \Psi^{'}(v), v - v_* \rangle \\
&= \langle F^{'}(v), v - x_* \rangle \geq F(v) - F_* \geq \frac{1}{c^2} \|v - v_*\|_2^2,
\end{aligned}$$

where the second inequality uses the convexity of $F$ and the last inequality uses the quadratic error bound condition (2). □

**Lemma 7.** *ssume that the problem (1) satisfies the QEB. Let $\bar{x} = \arg\min_{x \in \Omega} \langle \nabla f(\tilde{x}), x - \tilde{x} \rangle + \frac{L}{2}\|x - \tilde{x}\|_2^2 + \Psi(x)$. Then we have*

$$F(\bar{x}) - F_* \leq (L + L_f)^2 c^2 \|\bar{x} - \tilde{x}\|_2^2. \tag{29}$$

*Proof.* Let $x_*$ denote the closest optimal solution to $\bar{x}^{(s+1)}$. By Lemma 6 in the supplement, we have

$$\|\bar{x} - x_*\| \leq (L + L_f)c^2 \|\bar{x} - \tilde{x}\|.$$

By Lemma 5 in the supplement and the convexity of $F$, we have

$$F(\bar{x}) - F_* \leq -\langle F'(\bar{x}), x_* - \bar{x} \rangle \leq (L + L_f)\|\bar{x} - \tilde{x}\|\|\bar{x} - x_*\|.$$

Combining the two inequalities above together leads to

$$F(\bar{x}) - F_* \leq (L + L_f)^2 c^2 \|\bar{x} - \tilde{x}\|^2.$$

□