[Reviews · NeurIPS 2017]

Reviewer 1



The results of this paper appear to be interesting. Here are some comments: 1. To ensure the solution stays in a compact set, the author add a constraint R(x) \leq B. Then, authors have to use a sub-algorithm for evaluating the proximity operator of R under this constraint. This is a main drawback of this work and the author should address carefully the effect of inexact to rate of convergence. 2. The current state of this paper consider the case where \theta is in (0, 0.5). In some application, theta can be greater than 0.5. Can the authors provide some comments to explain why ? 3. More example for HEB condition should be provide. 2. The notation is overlap: R is the number of iteration and function.

Reviewer 2



The paper studies the SVRG algorithm (stochastic variance reduced gradient) under quadratic error bound condition and more generally under Holderian error bound condition. The main contribution of the paper is to adapt the error bound parameter without knowing it in advance, where the so called error bound parameter is a similar quantity as the usual strong convexity parameter. The idea of the paper is very interesting and I have checked the proof is correct. However, I find the presentation of the algorithm a bit obscure, it might be better to give more intuitions and explanations on it in the main paper instead of going into the details of the proof. For exemple, it would be nice to comment on the difference between the complexity before and after the restart strategy and compare it with the lower bound if possible. The main idea of the paper is to predefine the number of iterations R and T for the outer and inner loop operations and check whether a sufficient decrease condition is satisfied, if not the estimate error bound parameter is too small compare to its true value and we should increase R and T. In contrast to the algorithm 1 which can run until any accuracy, the adaptive method requires to set the target accuracy in advance, because R depend on \epsilon. Is there a way to remove such dependency and how does it set in the experiments? In the experiments, a flat plateau sometimes occurs, can authors comment on such observation? Overall, the theoretical result of the paper is solid with promising experimental evaluations but the presentation is improvable, a major revision of the text is needed. #EDIT Thank you for author's feedback. I keep my score because of the presentation is largely improvable.

Reviewer 3



This paper introduces a restarting scheme for SVRG specialised to the case where the underlying problem satisfies the quadratic error bound (QEB) condition, a weaker condition than strong convexity that still allows for linear convergence. The algorithm solves the problem of having to know the value c of the QEB constant before hand. The restarting scheme applies full SVRG repeatedly in a loop. If after the application of SVRG the resulting iterate is less than a 25% relative improvement (according to a particular notion of solution quality they state), the value of c is increased by \sqrt{2}, so that the next run of SVRG uses double the number of iterations in it's inner loop (T propto c^2). The use of doubling schemes for estimating constants in optimisation algorithms is a very standard technique. It's use with SVRG feels like only an incremental improvement. A particular problem with such techniques is setting the initial value of the constant. Too small a value will result in 5-10 wasted epochs, where as too large a value results in very slow convergence. The experiments section is well-explained, with standard test problems used. I do have some issues with the number of steps shown on the plots. The x axis #grad/n goes to 1000 on each, which is unrealistically large. It masks the performance of the algorithm during the early iterations. In the SVRG and SAGA papers the x axis spans from 0 to 30-100 of #grad/n, since this represents the range where the loss bottoms out on held-out tests sets. It's hard to tell from the plots as they are at the moment if the algorithm works well where it matters. This would be much clearer if error on a test set was shown as well. On the first plot, a comparison is made against SVRG with a fixed number of inner iterations T, for a variety of T. The largest T tried had the best performance, I would suggest adding larger values to the plot to make clear that using too large a value results in worse performance (i.e. there is a sweet spot). I would remove the T=100 from the plot to make room. In terms of language, the paper is ok, although in a number of places the language is a little informal. There are a moderate number of grammatical issues. The paper does require additional proof reading to get it up to standard.